# Nutrition Profile and Animal-Tested Safety of *Morchella esculenta* Mycelia Produced by Fermentation in Bioreactors

**DOI:** 10.3390/foods11101385

**Published:** 2022-05-11

**Authors:** I-Chen Li, Lynn-Huey Chiang, Szu-Yin Wu, Yang-Chia Shih, Chin-Chu Chen

**Affiliations:** 1Biotech Research Institute, Grape King Bio Ltd., Taoyuan City 325, Taiwan; ichen.li@grapeking.com.tw (I.-C.L.); lynnhuey.chiang@grapeking.com.tw (L.-H.C.); sy.wu@grapeking.com.tw (S.-Y.W.); 2Department of Biotechnology, Asia University, Taichung City 413, Taiwan; angelashih@asia.edu.tw; 3Department of Food Science, Nutrition, and Nutraceutical Biotechnology, Shih Chien University, Taipei City 104, Taiwan; 4Institute of Food Science and Technology, National Taiwan University, Taipei City 106, Taiwan; 5Department of Bioscience Technology, Chung Yuan Christian University, Taoyuan City 320, Taiwan

**Keywords:** *Morchella esculenta*, true morels, fermentation, safety, NOAEL

## Abstract

*Morchella esculenta* (ME), or “true” morel mushrooms, are one of the most expensive mushrooms. *M. esculenta* contain all the important nutrients including carbohydrates, proteins, polyunsaturated fatty acids, and several bioactive compounds such as polysaccharides, organic acids, polyphenolic compounds, and tocopherols, which are promising for antioxidant, immunomodulation, anti-cancer, and anti-inflammatory applications. However, the *M. esculenta* fruiting body is difficult to collect in nature and the quality is not always reliable. For this reason, the cultivation of its mycelia represents a useful alternative for large-scale production. However, for *M. esculenta* mycelia to be used as an innovative food ingredient, it is very important to prove it is safe for human consumption while providing high-quality nutrients. Hence, for the first time in this study, the nutritional composition, as well as 90 days of oral toxicity of fermented ME mycelia in Sprague Dawley rats, is examined. Results showed that the ME mycelia contained 4.20 ± 0.49% moisture, 0.32 ± 0.07% total ash, 17.17 ± 0.07% crude lipid, 39.35 ± 0.35% crude protein, 38.96 ± 4.60% carbohydrates, and 467.77 ± 0.21 kcal/100 g energy, which provides similar proportions of macronutrients as the U.S. Dietary Reference Intakes recommend. Moreover, forty male and female Sprague Dawley rats administrating ME mycelia at oral doses of 0, 1000, 2000, and 3000 mg/kg for 90 days showed no significant changes in mortality, clinical signs, body weight, ophthalmology, and urinalysis. Although there were alterations in hematological and biochemical parameters, organ weights, necropsy findings, and histological markers, they were not considered to be toxicologically significant. Hence, the results suggest that the no-observed-adverse-effects level (NOAEL) of ME mycelia was greater than 3000 mg/kg/day and can therefore be used safely as a novel food at the NOAEL.

## 1. Introduction

A species of fungus in the genus *Morchella* is generally called true morels and “yang du jun” in Mandarin (lamb’s stomach mushroom) as their outer appearance resembles lamb stomachs [1]. These mushrooms are widely found in temperate regions of the northern hemisphere, where they fruit for only a few weeks in the spring [1]. In Chinese culture, morels have a long history of human consumption as medicine to treat a variety of stomach problems, which was recorded in the ancient Chinese book Compendium of Materia Medica written by the Ming Dynasty physician Li Shi-Zhen in 1596 [2]. In Japan, Malaysia, India, and Pakistan, morels function as natural aphrodisiacs based on ethnomycological studies [3]. In current scientific research, morel mushrooms were found to contain profound anti-tumor [4], immunoregulatory [4], anti-inflammatory [5], and antioxidant activities [6]. As morels have enormous health benefits, the market demand for morels has increased accordingly. However, because morels have a complex life cycle, the cultivation of morel fruiting bodies remains unsuccessful, despite more than 100 years of effort [7]. It was only until recently that the first indoor cultivation of *M. rufobrunnea* in the United States, and *M. importuna*, *M. sextalata,* and *M. eximia* in China was reported [2]. Nevertheless, as morels have not yet been farmed successfully on a large scale, the industry still has to harvest wild mushrooms to meet the global demand. As such, the price of morel fruiting bodies remains high, and its dry product price is around USD 360 per kg [8]. Hence, research on *Morchella* mycelia and its use as a substitute for the natural fruiting bodies has received the attention of many investigators.

From 1883, scientists in the United States and France began to carry out experiments on morel production including strains such as *M. crassipes* and *M. esculenta* in submerged cultivation [9]. During that time, researchers focused on using their nutritional properties and aroma properties for the development of functional foods and flavoring agents, respectively [10]. Eventually, the first commercial product, “Morel mushroom flavoring”, produced by the special products division of Mid-America Dairymen Inc., was introduced to the market in the 1950s [11]. However, due to poor market response and availability of imported dried fruiting bodies at lower cost, the company stopped manufacturing and the research of morel mycelia stopped making progress. It was not until 20 years later, after an increased awareness of the health benefits of consuming mushroom products as a support to a healthier life, did the scientists begin to investigate these poorly understood fungi. According to recent studies, morels in submerged culture, especially in the *M. esculenta* strain, were shown to exhibit anti-cancer [12], anti-inflammatory [12], anti-microbial [13], hepatoprotective [14], and renal protective activities [15]. As *M. esculenta* mycelia have various health-promoting and medicinal properties, they could be used as raw materials in the formulation and development of new functional foods for health-conscious consumers. However, for *M. esculenta* mycelia to be used as an innovative food ingredient, it is very important to prove it is safe for human consumption while providing high-quality nutrients.

As mentioned, *M. esculenta* mycelia were fermented as early as the 1950s, and the product was approved by the US FDA [16]. However, the safety tests during that time were different from modern methods. Moreover, toxicological information regarding the chronic ingestion of *M. esculenta* mycelia is unavailable. Hence, to the best of our knowledge, this study is the first to examine the nutritional composition as well as 90 days of oral toxicity of fermented *M. esculenta* mycelia in Sprague Dawley rats to confirm their potential use as healthy food ingredients and dietary supplements.

## 2. Materials and Methods

### 2.1. Preparation of M. esculenta Mycelia Powder

The morel strain used in this experiment was isolated from fresh morel mushrooms purchased from the French market (Paris, France). The strain was later authenticated as *Morchella esculenta* (ME) by the Institute of Microbiology, Chinese Academy of Sciences, Beijing, China. In brief, ME was activated in a potato dextrose agar (PDA) and transferred as inoculum into a 2000 mL Erlenmeyer flask containing 1000 mL of fresh medium with 2% sucrose, 0.3% yeast extract, 2% soybean powder, 0.05% KH_2_PO_4_, and 0.05% MgSO_4_. Liquid cultures were incubated at 25 °C with 120 rpm shaking (FIRSTEK S103, Chuanhua Precision Corp., Taipei, Taiwan) for 5 days before their transfer to a 5-ton bioreactor for mass production. After a week of the fermentation process, the mycelia were harvested, freeze-dried, and milled to a fine powder.

### 2.2. Nutritional Analyses

Nutritional analysis was carried out according to the procedure of Association of Official Analytical Chemists (AOAC) for moisture (method 925.10), ash (method 923.03), crude fiber (method 962.09), crude fat (method 920.39), and crude protein content (method 979.06) [17]. The carbohydrate was calculated by subtracting the sum (g/100 g dry matter) of crude protein, crude fat, moisture, and ash from 100 g. The energy value was calculated by multiplying proteins, carbohydrates, and fats by 4, 4, and 9, respectively.

### 2.3. Animals and Ethics

Forty 5–6-week-old male and female Sprague Dawley rats were purchased from BioLASCO, Taipei, Taiwan and acclimated for a week before the treatment. These animals were housed in pairs in a room with a controlled temperature of 22 ± 3 °C, a humidity of 55 ± 15%, and a photoperiod of 12/12 h light–dark cycles. Standard laboratory diet (MFG, Oriental Yeast Co. Ltd., Tokyo, Japan) and sterile distilled water were provided ad libitum. This study was conducted according to the Organization for Economic Co-operation and Development (OECD) guidelines [18] and approved by the Institutional Animal Care and Use Committee (IACUC 106-9i; approval year 2017) of Super Laboratory in Taiwan.

### 2.4. Experimental Design

Rats were randomly assigned to minimize weight variation between groups and allocated into four groups, each containing 10 animals per sex. These animals were treated once daily by oral gavage administration using a straight, stainless-steel-ball-tipped feeding needle of either 0 (sterile distilled water as negative control), 1000 (low dose), 2000 (mid-dose), or 3000 (high dose) mg/kg bw/day ME in a 10 mL/kg application volume. The dose levels were determined based on the results of our previous teratotoxic assessment [19], which used the same method in this study to prepare ME samples and showed no adverse effect when up to 3000 mg/kg bw/day of ME was given to female pregnant rats during the major embryonic organogenesis period (days 6–15). Prior to treatment and weekly thereafter, all rats were observed once daily for clinical signs including morbidity as well as mortality and measured once weekly for body weight and food intake. At end of the 90-day study, the blood was withdrawn from the abdominal aorta and urine samples were collected in surviving animals fasted overnight. Finally, these rats were anesthetized with carbon dioxide, dissected, subjected to complete necropsy as well as gross pathological examination, and weighed before exsanguination.

### 2.5. Hematology, Clinical Chemistry, and Urinalysis

Blood samples were collected in EDTA tubes for the determination of hematological parameters and standard serum separator tubes for clinical chemistry. The 13 hematological parameters included white blood cells (WBC), red blood cells (RBC), hemoglobin, hematocrit, mean corpuscular volume (MCV), mean corpuscular hemoglobin (MCH), mean corpuscular hemoglobin concentration (MCHC), platelets, neutrophils, lymphocytes, monocytes, eosinophils, and basophils, which were examined using the XT-1800i Automated Hematology Analyzer (Sysmex, Kobe, Japan). Prothrombin time (PT) in plasma was measured with a CA-1500 Automated Blood Coagulation Analyzer (Siemens Healthineers, Erlangen, Germany). The 18 clinical biochemistry parameters, such as glucose, blood urea nitrogen (BUN), creatinine, alanine aminotransferase (ALT), aspartate aminotransferase (AST), total protein (TP), albumin, alkaline phosphatase (ALP), gamma-glutamyl transpeptidase (ɣ-GT), cholesterol, triglyceride, calcium, phosphorus, sodium, potassium, chloride, globulin, and total bilirubin, were analyzed using a 7070 autoanalyzer (Hitachi, Tokyo, Japan). Urinalysis was conducted using PU 4010 Compact Urine Analyzer (Arkray, Kyoto, Japan) for parameters including glucose, bilirubin, ketone body, specific gravity, pH, protein, urobilinogen, nitrite, occult blood, and leukocyte esterase. The remaining urine was centrifuged, and the following parameters were evaluated using a microscope: RBC, WBC, epithelial cell, and crystals.

### 2.6. Necropsy, Organ Weights, and Histopathology

At necropsy, a careful examination of the external body surface including all orifices, the cranial, thoracic, abdominal cavities, and their contents was performed. Organs such as brain, lungs, heart, kidneys, liver, spleen, adrenal glands, testes (fixed in modified Davidson’s fluid), as well as ovaries, were excised, weighed, and fixed in 10% neutral formalin for 24 h. Additional tissues including the aorta, bone marrow, duodenum, jejunum, ileum, caecum, colon, rectum, eyes, esophagus, mammary gland, Harderian gland, trachea, lung, lymph node, pancreas, sciatic nerve, pituitary, prostate gland, salivary gland, skin, spinal cord, stomach, thigh muscle, thymus, thyroid/parathyroid gland, urinary bladder, and uterus were also fixed in 10% neutral formalin. After a fixative for 24 h, these samples were trimmed, embedded in paraffin, cut to a thickness of approximately 2–5 µm, stained with hematoxylin and eosin (H&E), and examined by light microscopy. Histopathology was performed on all organs and tissues from the high-dose and the control groups. Only if histopathological changes were identified in the high-dose group were the lower-dose groups examined.

### 2.7. Statistical Analysis

All values were presented as mean ± SD. Statistical differences were evaluated by one-way ANOVA followed by Duncan’s multiple range test using SPSS Statistics v22 (IBM Corp., Armonk, NY, USA). The level of significance was taken as *p* < 0.05.

## 3. Results

### 3.1. Nutritional Analysis of ME Mycelia

The nutritional analysis of the ME mycelia is shown in Table 1.

### 3.2. Effect of ME Mycelia on Body Weight and Food Consumption

No morbidity or mortality was found in any rats during the 90-day oral administration of ME mycelia. There were no significant differences in body weight among groups (*p* > 0.05; Figure 1). Similarly, no significant difference was found in food consumption between ME-mycelia-treated groups and the corresponding control groups in females (*p* > 0.05; Figure 2). However, a significant decrease in chow intake was observed for high-dose male rats at weeks 6, 9, 11, and 12 (*p* < 0.05; Figure 2). Nevertheless, this decrease was not dose-dependent and was only observed in male rats. Therefore, such alterations do not represent treatment-related toxicological effects and were considered to be incidental signs.

### 3.3. Effect of ME Mycelia on Hematology, Clinical Chemistry, and Urinalysis

90 days of ME mycelia treatment did not cause significant changes in the majority of the parameters in male and female rats at all the doses tested (*p* > 0.05; Table 2). However, statistical differences in hemoglobin values were observed in the male mid- and high-dose-treated groups when compared to the control group (*p* < 0.05; Table 2). In addition, in the females, the high-dose-treated group has an increased MCHC value and a decreased PT time in comparison to the control group (*p* < 0.05; Table 2). Nonetheless, these differences were determined to be unrelated to the treatment because the alterations were observed in one sex and no dose–response relationship was detected.

The terminal clinical chemistry for female and male rats is shown in Table 3. For most parameters, no significant differences were found between the administration groups and the control group during the 90 days of treatment (*p* > 0.05; Table 3). However, the following changes relative to the vehicle control group were observed in the males: higher ALT values in the 3000 mg/kg dosing group; higher calcium values in the 1000 mg/kg dosing group; higher phosphorus values in the 2000 and 3000 mg/kg dosing groups and higher sodium values in the 1000 mg/kg dosing group (*p* < 0.05; Table 3). Moreover, a comparatively low BUN value and low ALT value were observed in the female 1000 and 2000 mg/kg dosing group, respectively. As these values all fall within the normal range (including our historical control as well as reference data [21]; ALT = 18.72–85.05 U/L; calcium = 11.20–13.02 mg/dL; phosphorus = 9.49–13.01 mg/dL; sodium = 142.47–150.41 meq/L and BUN = 11.8–20.0 mg/dL) and are not dose-dependent; these discrepancies were considered to be normal biological variations. 

The urine volume, urine color, urine clarity, glucose, bilirubin, ketone body, specific gravity, pH, protein, urobilinogen, nitrite, occult blood, and leukocyte esterase of the control group and the treatment groups were similar to each other (Table 4). Furthermore, the analysis of the RBC, WBC, epithelial cell, and crystals in the urine sediments showed that there was no significant difference found across all groups (*p* > 0.05; Table 4).

### 3.4. Effect of ME Mycelia on Necropsy, Organ Weights, and Histopathology

At the necropsy, there were no treatment-related gross findings in any of the animals. For absolute organ weight in the treatment group, there were significant decreases in the weight of the heart and the spleen of high-dose-treated males and mid-dose-treated females, respectively (*p* < 0.05; Table 5). However, these changes did not resemble those of relative organ weights (Table 6). Additionally, the relative ovary weight in the female 1000 mg/kg dosing group was higher than that in the control group. Nonetheless, the differences observed were considered to be random and not related to ME mycelia because they were within normal biological variability (including our historical control as well as reference data [22]; male absolute heart weight: 1.544–1.916 g; female absolute spleen weight: 0.43–0.78 g; and relative ovary weight: 0.024–0.044 g) and without any dose–response relationship.

For further histopathological examination, the following lesions were observed in the high-dose treatment groups of both sexes: cysts in the pituitary gland, minimal mononuclear cell infiltration in the harderian gland, minimal adipocyte infiltration in the pancreas, minimal cytoplasmic vacuolation as well as minimal inflammatory cell infiltration in the liver, minimal cytoplasmic vacuolation in the adrenal gland, minimal mononuclear cell infiltration as well as a cyst in the thyroid gland, minimal mononuclear cell infiltration in the heart, minimal hemorrhage in the thymus, minimal hemorrhage in the lung, minimal cysts in the kidney, minimally increased adipocytes in the femur, and minimal mononuclear cell infiltration in the prostate gland (data not shown). According to a previous study, old rats will have a subset of lesions as part of the progressive decline in organ function [23]. As the frequency of these occurrences did not differ significantly from those of the vehicle control group (*p* > 0.05), these lesions were considered sporadic and not to be associated with the ME mycelia.

## 4. Discussion

When the fruiting body of morel is mixed with rice and vegetables, it is as nutritious as fish or meat [24]. According to a recent study, *M. esculenta* contained all the important nutrients including carbohydrates, proteins, polyunsaturated fatty acids, and several bioactive compounds such as polysaccharides, organic acids, polyphenolic compounds, and tocopherols [25]. These substances, when extracted from *M. esculenta*, are promising for antioxidant, immunomodulation, anti-cancer, and anti-inflammatory applications [26]. However, collecting *M. esculenta* is a difficult job as it can only be collected from March to June [27] and is mostly found in areas burned by wildfire [28]. Moreover, the cultivation of *M. esculenta* fruiting bodies takes a long time, requires a large volume of substrates and space, and it is difficult to control the product quality. As there is a great need to regularly supply the market with high-quality *M. esculenta*, the submerged culture of *M. esculenta* appears to be a promising alternative method for the production.

For the use of ME mycelia, it is necessary to prove that it possesses nutritional values comparable to those of mushroom fruiting bodies. A recent study has shown that the fresh weight of *M. esculenta* fruiting body contained 89.41 ± 1.73% moisture and its dry weight consisted of 7.81 ± 0.36% ash, 11.31 ± 0.64% crude protein, 2.54 ± 0.22% fat, 78.33 ± 1.01% total carbohydrates, as well as 1618.04 ± 3.04 kJ total energy [29]. In this study, the mycelia contained 4.20 ± 0.49% moisture, 0.32 ± 0.07% total ash, 17.17 ± 0.07% crude lipid, 39.35 ± 0.35% crude protein, 38.96 ± 4.60% carbohydrates, and 467.77 ± 0.21 kcal/100 g energy, which were in agreement with the results reported previously [30]. Apart from the moisture content that is known to be higher in fresh fruiting bodies, the ash and carbohydrate contents were also higher in the fruiting body, while the protein and the fat content of the mycelium were higher than those of the fruiting body. When compared to 19 other studied culinary–medicinal mushroom mycelia in another study, the fermented *M. esculenta* mycelia in this study has the highest protein content while the carbohydrate, fat, and ash contents were lower when compared to mycelia of *Ganoderma tsugae*, *Grifola frondosa*, *Hypsizygus marmoreus* white, and *Pleurotus citrinopileatus* [30].

Previous studies have shown that *M. esculenta* fruiting bodies contained high levels of protein and possessed a variety of essential amino acids including glutamic acid, aspartic acid, and leucine that are essential for humans [31]. Moreover, the protein hydrolysate in *M. esculenta* mushrooms was shown to present strong antioxidant abilities [32]. As fermented *M. esculenta* mycelia had a higher amount of protein than fruiting bodies, it suggested that the fermented *M. esculenta* mycelia may be a better source of protein supplement with stronger antioxidant activity. However, further investigation is required.

Mushrooms are also healthy sources of essential fatty acids such as linoleic, oleic, and linolenic, which cannot be directly synthesized in the human body but are required for health [33]. Compared to the fat composition of 39 common and popular mushroom species that ranged from 0.2 to 8% [34], the fermented *M. esculenta* mycelia in this study had 17.17% crude fat, which is not typical for fruiting bodies. After subdividing into saturated and unsaturated fats, there was 14.6% saturated fat, 0% trans-fat, and 85.4% unsaturated fats in the fermented *M. esculenta* mycelia (data not shown). Consistent with a previous study, unsaturated fatty acids were higher than saturated fatty acids, and linoleic, oleic, and palmitic were the predominant fatty acids in the lipid content of *M. esculenta* [25]. According to a previous study, unsaturated fats are an important part of a healthy diet as these fats help reduce the risk of heart disease and lower cholesterol levels when they replace saturated fats in the diet [35]. As human metabolism cannot synthesize linoleic and linolenic acids as they lack enzymes for ω-3 desaturation [36], this suggested that fermented *M. esculenta* mycelia can be an important source of essential fatty acids for a healthy human diet.

As the nutritional profile of *M. esculenta* mycelium was not inferior to that of wild *M. esculenta* fruiting bodies, this suggested that fermented *M. esculenta* mycelium may be a better alternative for healthcare product development. However, mushroom mycelium and fruiting bodies are two distinct parts of the fungal organism; hence, it is important to take its safety and edibility as a food ingredient into consideration using a 90-day oral subchronic toxicity test. As studies evaluating the toxicity of ME mycelia have not been reported, the current study is the first to investigate the long-term effects of ME mycelia on overall health, behavior, organ function, hematology, clinical chemistry, and urinalysis in accordance with the OECD 408 procedures. During the course of the experiment, rats in all treatment groups and the control group all survived in good health. Despite the high-dose male group having a significant decrease in chow intake at weeks 6, 9, 11, and 12, it was back to normal in the next timeframe. This significant decrease may be caused by the 39.35% crude protein within the ME mycelia. According to previous studies, protein-rich loads can contribute to longer-lasting satiety, decrease subsequent energy intake, and lead to lower body weight [37]. Since it did not significantly affect the body weight and no abnormal behavior was recorded, it seems unlikely that this incident was related to the administration of ME mycelia.

As for the hematological examination, there were no significant differences among the treatment groups. A significant increase in hemoglobin values and a similar trend for the RBCs, however, were found in the male mid- and high-dose-treated groups, which may be due to the strong antioxidant effect of ME mycelia to prevent the destruction of RBCs from free radical formation [38]. In addition, a significant increase in the MCHC value and a decreased PT time were observed in the female high-dose-treated group when compared to the control group. Previous studies suggested that the increase in MCHC is usually associated with hemolysis [39]. However, no histological lesions, organ weight changes, and gross findings were observed in the relevant organs, indicating these changes were normal biological variations and are not related to the administration of ME mycelia.

In terms of the clinical chemical examination, in male rats, a higher ALT value in the 3000 mg/kg dosing group, a higher calcium value in the 1000 mg/kg dosing group, a higher phosphorus value in the 2000 and 3000 mg/kg dosing groups, and a higher sodium value in the 1000 mg/kg dosing group were observed. On the other hand, a comparatively low BUN value and low ALT value were observed in the female 1000 and 2000 mg/kg dosing groups, respectively. Both male and female rats administered with low-dose and medium-dose ME mycelia treatment exhibited decreased AST and ALT, suggesting that the lower doses of ME mycelia showed obvious protective effects on liver function. Although elevated serum ALT pointed to a potential risk of liver injury [40], the changes that occurred at the clinical chemistry levels did not correspond in the histopathological examination, indicating that ME mycelia had little effect on tissue levels. Nonetheless, these differences were sporadic, non-dose-dependent, and were not consistently observed between the sexes. Moreover, as these differences were small in magnitude, none were considered to be biologically or toxicologically significant.

In the analysis of organ weight and histopathology, although significant differences in absolute weights were discovered in the heart of high-dose-treated males and the spleen of mid-dose-treated females, these weights are within the normal range [22]. Moreover, no obvious lesions were found in further histopathological examination; hence, these changes could be considered of no clinical significance. In addition, a statistically significant increase in relative ovary weight was observed in female rats treated with 1000 mg/kg ME mycelia. However, there were no histopathological findings for the ovaries or uterus. Moreover, a relationship between absolute weight and relative weight was not found. Therefore, these changes were considered to be normal biological variations.

## 5. Conclusions

In conclusion, the results of the present 90-day repeated oral dose toxicity study demonstrated that administering ME mycelia does not exhibit toxicity in male and female rats under the experimental conditions adopted in the present study. Hence, the no-observed-adverse-effects level (NOAEL) for this study was greater than the dose of 3000 mg/kg/day. As a novel food, ME mycelia can be used safely at its NOAEL. However, for future study, a clinical trial of ME mycelia is needed to determine its safety for consumption by humans.

## Figures and Tables

**Figure 1 foods-11-01385-f001:**
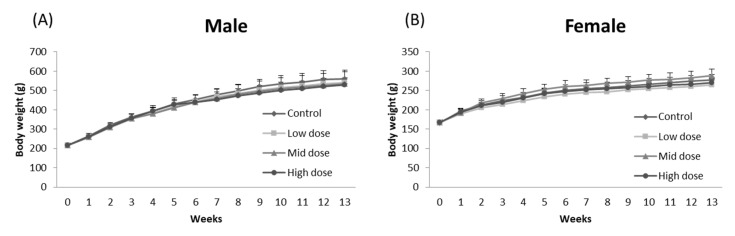
Effect of ME mycelia on the average body weight of (**A**) male and (**B**) female rats throughout the experiment. Data expressed as mean ± SD (*n* = 10).

**Figure 2 foods-11-01385-f002:**
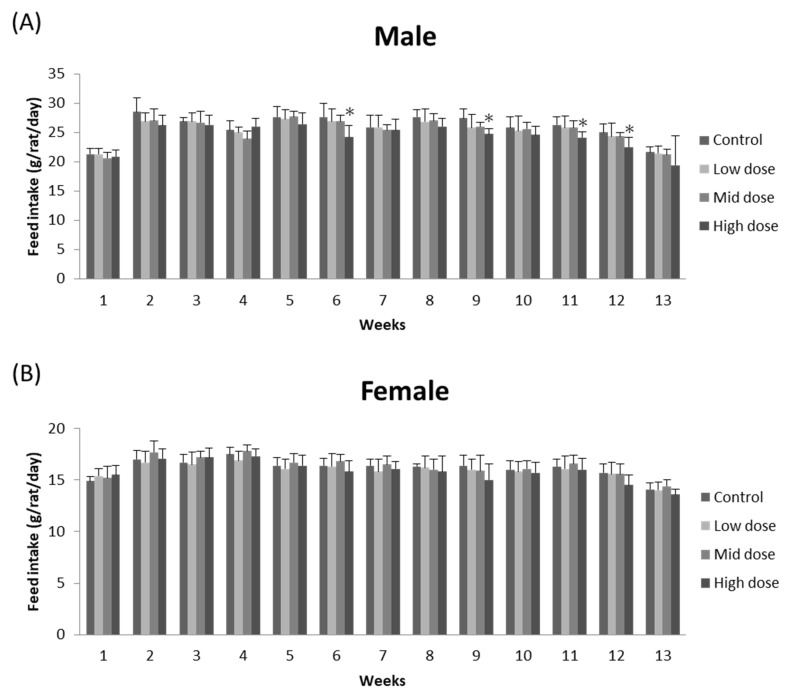
Effect of ME mycelia on the average food consumption of (**A**) male and (**B**) female rats throughout the experiment. Data expressed as mean ± SD (*n* = 10). * indicates significance (*p* < 0.05).

**Table 1 foods-11-01385-t001:** Nutritional analysis of ME powder.

Nutritional	(%)			Population Norms (%) ^a^
Moisture	4.20	±	0.49	
Ash	0.32	±	0.07	
Crude lipid	17.17	±	0.07	20–35
Crude protein	39.35	±	0.35	10–30
Carbohydrate	38.96	±	4.60	40–65
Energy (kcal/100 g)	467.77	±	0.21	

Each value is expressed in mean ± SD (*n* = 2). ^a^ According to the U.S. Recommended Dietary Reference Intakes [20].

**Table 2 foods-11-01385-t002:** Effect of ME mycelia on hematology of male and female rats.

	Control			Low			Mid			High		
ME Mycelia (mg/kg)	0			1000			2000			3000		
Parameters												
Male												
WBC (10^3^/μL)	11.4	±	2.0	12.0	±	2.7	11.3	±	1.9	12.4	±	2.4
RBC (10^6^/μL)	9.3	±	0.2	9.7	±	0.5	9.7	±	0.8	9.8	±	0.4
Hemoglobin (g/dL)	16.4	±	0.6	17.1	±	1.1	17.4	±	1.3 *	17.4	±	0.8 *
Hematocrit (%)	46.8	±	1.2	47.6	±	1.7	47.7	±	1.4	48.0	±	1.0
MCV (fL)	49.5	±	1.1	49.9	±	1.8	49.7	±	1.5	49.3	±	1.0
MCH (pg)	17.6	±	0.4	17.7	±	0.7	17.9	±	0.5	17.7	±	0.4
MCHC (g/dL)	35.5	±	0.5	35.4	±	0.4	35.7	±	0.4	35.7	±	0.5
Platelet (10^3^/μL)	1054.1	±	113.9	1103.2	±	133.3	1116.5	±	146.0	1169.0	±	177.1
Neutrophil (%)	20.8	±	3.4	22.7	±	7.9	20.9	±	6.4	19.7	±	4.7
Lymphocyte (%)	72.5	±	2.9	71.1	±	8.3	73.1	±	6.9	74.5	±	4.6
Monocyte (%)	5.3	±	0.8	5.0	±	1.2	4.8	±	1.0	4.8	±	0.8
Eosinophil (%)	1.3	±	0.4	1.1	±	0.3	1.1	±	0.4	0.9	±	0.4
Basophil (%)	0.1	±	0.1	0.2	±	0.1	0.2	±	0.1	0.2	±	0.1
PT (s)	12.2	±	1.3	11.7	±	0.4	11.6	±	0.8	12.6	±	1.8
Female												
WBC (10^3^/μL)	8.8	±	2.1	8.4	±	1.7	8.9	±	1.6	9.3	±	1.4
RBC (10^6^/μL)	8.8	±	0.8	8.8	±	0.7	8.7	±	0.7	8.8	±	0.4
Hemoglobin (g/dL)	15.9	±	0.9	16.4	±	1.2	16.0	±	1.1	16.2	±	0.6
Hematocrit (%)	45.8	±	3.0	45.8	±	2.8	45.4	±	3.2	45.4	±	2.0
MCV (fL)	51.9	±	1.7	52.4	±	2.1	52.4	±	2.2	51.7	±	1.9
MCH (pg)	18.5	±	0.6	18.7	±	0.5	18.4	±	0.5	18.5	±	0.7
MCHC (g/dL)	34.7	±	1.4	35.0	±	1.0	35.0	±	0.6	35.8	±	0.3 *
Platelet (10^3^/μL)	1088.6	±	145.4	1052.9	±	114.8	1072.1	±	96.2	1038.2	±	100.5
Neutrophil (%)	14.0	±	3.1	14.8	±	5.1	13.2	±	5.2	17.3	±	4.5
Lymphocyte (%)	80.3	±	3.5	79.2	±	6.0	81.6	±	5.2	77.6	±	5.0
Monocyte (%)	4.1	±	1.0	4.5	±	1.7	3.8	±	1.0	3.9	±	1.1
Eosinophil (%)	1.4	±	0.6	1.3	±	0.6	1.2	±	0.3	1.1	±	0.4
Basophil (%)	0.2	±	0.1	0.2	±	0.1	0.2	±	0.2	0.2	±	0.1
PT (s)	9.9	±	0.2	9.8	±	0.1	9.8	±	0.1	9.6	±	0.2 *

Data are expressed as mean ± SD (*n* = 10). White blood cell (WBC); Red blood cell (RBC); Mean corpuscular volume (MCV); Mean corpuscular hemoglobin (MCH); Mean corpuscular hemoglobin concentration (MCHC); prothrombin time (PT). * indicates significance (*p* < 0.05).

**Table 3 foods-11-01385-t003:** Effect of ME mycelia on clinical chemistry of male and female rats.

	Control			Low			Mid			High		
ME Mycelia (mg/kg)	0			1000			2000			3000		
Parameters												
Male												
Glucose (mg/dL)	258.7	±	33.4	287.4	±	60.2	256.4	±	51.4	276.8	±	63.5
BUN (mg/dL)	12.1	±	0.9	11.8	±	1.4	12.1	±	1.4	11.9	±	1.4
Creatinine (mg/dL)	0.5	±	0.1	0.5	±	0.0	0.5	±	0.1	0.5	±	0.1
AST (U/L)	130.3	±	37.9	100.3	±	20.9	107.0	±	36.4	134.7	±	48.3
ALT (U/L)	50.2	±	32.4	42.3	±	12.0	39.6	±	16.2	60.9	±	28.6 *
Total protein (g/dL)	6.6	±	0.3	6.7	±	0.3	6.5	±	0.3	6.5	±	0.4
Albumin (g/dL)	3.5	±	0.1	3.6	±	0.1	3.5	±	0.1	3.5	±	0.3
ALP (U/L)	79.4	±	15.0	90.6	±	20.2	76.4	±	16.4	80.2	±	23.0
γ-GT (U/L)	<2.0			<2.0			<2.0			<2.0		
Cholesterol (mg/dL)	30.5	±	6.1	27.9	±	5.9	33.9	±	9.5	24.1	±	5.1
Triglyceride (mg/dL)	67.3	±	18.9	58.7	±	18.1	58.1	±	15.2	51.5	±	14.1
Calcium (mg/dL)	11.8	±	0.3	12.5	±	0.6 *	12.0	±	0.4	12.0	±	0.6
Phosphorus (mg/dL)	11.4	±	0.9	12.0	±	0.6	12.6	±	1.5 *	12.7	±	0.9 *
Sodium (meq/L)	146.7	±	1.5	148.5	±	1.8 *	147.8	±	1.0	146.2	±	1.9
Potassium (meq/L)	9.0	±	0.9	8.4	±	1.2	9.3	±	2.4	9.8	±	1.3
Chloride (meq/L)	96.8	±	1.5	97.0	±	1.3	97.6	±	1.1	97.1	±	1.7
Globulin (g/dL)	3.1	±	0.2	3.1	±	0.2	3.0	±	0.2	3.0	±	0.2
Total bilirubin (mg/dL)	<0.04			<0.04			<0.04			<0.04		
Female												
Glucose (mg/dL)	150.0	±	37.1	138.5	±	39.4	138.0	±	36.8	129.1	±	26.9
BUN (mg/dL)	13.4	±	1.5	13.7	±	1.5	13.6	±	1.7	12.0	±	1.0 *
Creatinine (mg/dL)	0.5	±	0.0	0.5	±	0.1	0.5	±	0.1	0.5	±	0.0
AST (U/L)	108.6	±	26.9	103.6	±	33.5	92.4	±	13.6	118.6	±	22.6
ALT (U/L)	43.3	±	20.7	34.9	±	22.1	23.6	±	4.0 *	33.8	±	7.3
Total protein (g/dL)	7.3	±	0.7	7.2	±	0.4	7.0	±	0.4	7.2	±	0.2
Albumin (g/dL)	4.3	±	0.4	4.3	±	0.2	4.0	±	0.3	4.1	±	0.2
ALP (U/L)	37.8	±	7.8	17.0	±	14.7	45.5	±	13.7	37.0	±	9.5
γ-GT (U/L)	<2.0			<2.0			<2.0			<2.0		
Cholesterol (mg/dL)	44.7	±	10.6	44.5	±	8.6	45.4	±	8.2	51.6	±	11.4
Triglyceride (mg/dL)	47.0	±	4.7	41.7	±	4.3	41.8	±	11.6	42.1	±	5.7
Calcium (mg/dL)	11.9	±	0.6	12.1	±	0.6	11.8	±	0.4	11.9	±	0.4
Phosphorus (mg/dL)	10.7	±	1.1	10.8	±	0.8	11.3	±	1.2	11.3	±	0.5
Sodium (meq/L)	145.2	±	2.7	144.6	±	1.7	144.1	±	1.0	143.6	±	2.0
Potassium (meq/L)	10.9	±	2.5	10.5	±	1.8	10.0	±	0.9	11.0	±	1.4
Chloride (meq/L)	97.8	±	2.5	97.3	±	1.3	96.5	±	1.2	96.5	±	1.5
Globulin (g/dL)	3.0	±	0.4	2.9	±	0.2	3.0	±	0.2	3.1	±	0.1
Total bilirubin (mg/dL)	<0.04			<0.04			<0.04			<0.04		

Data are expressed as mean ± SD (*n* = 10). Blood urea nitrogen (BUN); Aspartate aminotransferase (AST); Alanine aminotransferase (ALT); Alkaline phosphatase (ALP); Gamma-glutamyl transferase (γ-GT). * indicates significance (*p* < 0.05).

**Table 4 foods-11-01385-t004:** Effect of ME mycelia on urinalysis of male and female rats.

		Control			Low			Mid			High		
ME Mycelia (mg/kg)		0			1000			2000			3000		
Parameters													
Male													
Volume (mL)		19.6	±	7.9	23.2	±	9.6	21.8	±	8.9	20.6	±	8.0
Color	Yellow	6			7			9			5		
	Pale yellow	4			3			1			5		
Clarity	Clear	6			4			6			6		
	Light turbid	3			2			3			3		
	turbid	1			4			1			1		
Glucose	Negative	10			10			10			10		
	Trace	0			0			0			0		
	500 mg/dL	0			0			0			0		
Bilirubin	Negative	10			10			10			10		
	1+	0			0			0			0		
	2+	0			0			0			0		
	3+	0			0			0			0		
Ketone body	Negative	4			7			5			8		
	Trace	5			3			4			2		
	≥15 mg/dL	1			0			1			0		
Specific gravity	≤1.005	0			0			1			0		
	1.005–1.030	9			10			9			9		
	≥1.030	1			0			0			1		
pH	≤6	0			0			0			0		
	6.0–8.0	10			10			10			10		
	≥8.0	0			0			0			0		
Protein	Negative	0			0			1			0		
	15 mg/dL	3			4			2			2		
	30 mg/dL	5			6			6			6		
	100 mg/dL	2			0			1			2		
Urobiliongen	Normal	7			8			7			9		
	≥2.0	3			2			3			1		
Nitrite	Negative	10			10			10			10		
	1+	0			0			0			0		
	2+	0			0			0			0		
Occult blood	Negative	9			10			10			10		
	Trace	1			0			0			0		
Leukocyte esterase	Negative	1			1			1			1		
	25 mg/dL	1			1			0			3		
	75 mg/dL	2			3			3			1		
	250 mg/dL	4			3			5			2		
	500 mg/dL	2			2			1			3		
RBC	0–1 hpf	10			10			10			10		
	2–5 hpf	0			0			0			0		
	6–10 hpf	0			0			0			0		
WBC	0–1 hpf	4			5			5			5		
	2–5 hpf	6			5			5			5		
	6–10 hpf	0			0			0			0		
EP	0–1 hpf	7			8			10			7		
	2–5 hpf	3			2			0			3		
	6–10 hpf	0			0			0			0		
	11–15 hpf	0			0			0			0		
Crystals	None found	0			1			2			0		
	Triple phosphates	10			9			8			10		
	Uric acid	0			0			0			0		
Female													
Volume (mL)		18.0	±	10.3	16.6	±	7.2	19.0	±	6.6	21.0	±	10.8
Color	Yellow	6			6			7			8		
	Pale yellow	4			4			3			2		
Clarity	Clear	5			5			6			6		
	Light turbid	2			3			3			3		
	Turbid	3			2			1			1		
Glucose	Negative	10			10			10			10		
	Trace	0			0			0			0		
	500 mg/dL	0			0			0			0		
Bilirubin	Negative	10			10			10			10		
	1+	0			0			0			0		
	2+	0			0			0			0		
	3+	0			0			0			0		
Ketone body	Negative	8			8			9			9		
	Trace	2			2			1			1		
	≥15 mg/dL	0			0			0			0		
Specific gravity	≤1.005	0			0			1			0		
	1.005–1.030	10			10			9			10		
	≥1.030	0			0			0			0		
pH	≤6	0			0			0			0		
	6.0–8.0	10			10			10			10		
	≥8.0	0			0			0			0		
Protein	Negative	7			7			9			8		
	15 mg/dL	1			3			1			2		
	30 mg/dL	1			0			0			0		
	100 mg/dL	1			0			0			0		
Urobiliongen	Normal	10			10			10			10		
	≥2.0	0			0			0			0		
Nitrite	Negative	7			7			6			7		
	1+	2			2			2			3		
	2+	1			1			2			0		
Occult blood	Negative	8			10			10			9		
	Trace	2			0			0			1		
Leukocyte esterase	Negative	7			8			10			10		
	25 mg/dL	0			2			0			0		
	75 mg/dL	1			0			0			0		
	250 mg/dL	1			0			0			0		
	500 mg/dL	1			0			0			0		
RBC ^†^	0–1 hpf	10			10			10			10		
	2–5 hpf	0			0			0			0		
	6–10 hpf	0			0			0			0		
WBC ^†^	0–1 hpf	8			10			10			10		
	2–5 hpf	2			0			0			0		
	6–10 hpf	0			0			0			0		
EP ^†^	0–1 hpf	5			6			3			3		
	2–5 hpf	5			4			7			7		
	6–10 hpf	0			0			0			0		
	11–15 hpf	0			0			0			0		
Crystals ^†^	None found	0			3			5			0		
	Triple phosphates	10			7			5			10		
	Uric acid	0			0			0			0		

Data are shown as the number of animals with signs among 10 animals observed. ^†^: Sediment.

**Table 5 foods-11-01385-t005:** Effect of ME mycelia on absolute organ weight (g) of male and female rats.

	Control			Low			Mid			High		
ME Mycelia (mg/kg)	0			1000			2000			3000		
Parameters												
Male												
Testes	3.397	±	0.24	3.475	±	0.31	3.520	±	0.224	3.44	±	0.304
Adrenal Glands	0.065	±	0.01	0.065	±	0.01	0.068	±	0.010	0.06	±	0.012
Spleen	0.780	±	0.070	0.779	±	0.180	0.780	±	0.076	0.77	±	0.144
Kidney	3.956	±	0.41	4.083	±	0.46	4.053	±	0.384	4.07	±	0.292
Heart	1.846	±	0.16	1.710	±	0.220	1.759	±	0.094	1.69	±	0.112 *
Brain	2.231	±	0.09	2.224	±	0.12	2.186	±	0.118	2.17	±	0.127
Liver	16.56	±	2.01	15.99	±	2.57	16.03	±	1.414	15	±	1.585
Female												
Ovary	0.078	±	0.01	0.090	±	0.02	0.077	±	0.005	0.08	±	0.023
Adrenal Glands	0.066	±	0.010	0.070	±	0.01	0.069	±	0.011	0.070	±	0.011
Spleen	0.452	±	0.08	0.467	±	0.040	0.520	±	0.074 *	0.480	±	0.058
Kidney	2.042	±	0.17	2.047	±	0.200	2.093	±	0.158	1.9	±	0.062
Heart	0.982	±	0.070	1.020	±	0.24	0.965	±	0.063	0.99	±	0.113
Brain	1.957	±	0.08	1.964	±	0.080	1.967	±	0.119	1.900	±	0.057
Liver	8.167	±	0.56	8.089	±	0.94	8.234	±	0.438	8.19	±	0.533

Data are expressed as mean ± SD (*n* = 10). * indicates significance (*p* < 0.05).

**Table 6 foods-11-01385-t006:** Effect of ME mycelia on relative organ weight (g) of male and female rats.

	Control			Low			Mid			High		
ME Mycelia (mg/kg)	0			1000			2000			3000		
Parameters												
Male												
Testes	0.649	±	0.065	0.682	±	0.077	0.707	±	0.063	0.702	±	0.049
Adrenal Glands	0.012	±	0.003	0.013	±	0.003	0.014	±	0.003	0.013	±	0.003
Spleen	0.150	±	0.023	0.150	±	0.021	0.156	±	0.015	0.156	±	0.026
Kidney	0.755	±	0.083	0.798	±	0.087	0.814	±	0.088	0.829	±	0.045
Heart	0.352	±	0.034	0.333	±	0.026	0.353	±	0.023	0.347	±	0.031
Brain	0.427	±	0.048	0.437	±	0.049	0.439	±	0.040	0.444	±	0.023
Liver	3.144	±	0.230	3.101	±	0.276	3.212	±	0.255	3.050	±	0.249
Female												
Ovary	0.031	±	0.004	0.037	±	0.005 *	0.029	±	0.003	0.035	±	0.010
Adrenal Glands	0.026	±	0.004	0.029	±	0.004	0.026	±	0.004	0.029	±	0.004
Spleen	0.175	±	0.024	0.193	±	0.015	0.197	±	0.035	0.200	±	0.030
Kidney	0.798	±	0.084	0.847	±	0.070	0.791	±	0.084	0.790	±	0.040
Heart	0.383	±	0.034	0.423	±	0.102	0.364	±	0.025	0.412	±	0.063
Brain	0.764	±	0.053	0.814	±	0.048	0.743	±	0.077	0.791	±	0.045
Liver	3.188	±	0.285	3.341	±	0.282	3.104	±	0.215	3.411	±	0.311

Data are expressed as mean ± SD (*n* = 10). * indicates significance (*p* < 0.05).

## Data Availability

The data presented in this study are available on request from the corresponding author.

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
