# Peer review of "Nutrition Profile and Animal-Tested Safety of Morchella esculenta Mycelia Produced by Fermentation in Bioreactors"

_foods, 2022, doi:10.3390/foods11101385_

Round 1

Reviewer 1 Report

Although the study "Nutrition and safety of Morchella esculenta mycelia produced by fermentation in bioreactors" presents an important and interesting topic it needs major changes prior to publishing. Its major flaws are inadequate introduction (background/perspective) and lack of proper discussion. Minor English corrections should be done as well. Noone eats mycelia as a substitute for mushrooms (fruit body) due to its lack of flavor. Also, do we know the price of mycelia (amount which would be enough for one proper meal) in comparison with fruit bodies?

  1. The title should be changed to a more precise one: Nutritional profile and animal-tested safety  of Morchella esculenta mycelia produced by fermentation in bioreactor
  2. line 16: the cultivation
  3. line 18: "to prove its use" is too vague, make it more clear, for example, to prove it is safe for human consumption while providing high-quality nutrients.
  4. line 19: of oral toxicity
  5. line 24: delete one "rats"
  6. lines 40-42: It seems that the appreciation of morels justifies the need for their cultivation. The authors are suggesting mycelium cultivation. The final product-mycelia, is completely different from the sensory aspect and people praise them mainly because of their aroma and flavor (if mushrooms are taken as food); just consider the truffles! So, if high price and demand are to be the reason for an alternative source of this food then the real reason for their appreciation must be taken into account. Thus, this part of your paper needs to be rewritten from the nutritive perspective or medicinal one, not flavor and delicacy-based. Mycelium lacks any of that. Or, if this stance is to be kept the research needs sensory evaluation of bioreactor obtained mycelia.
  7. lines 68-70: the sample as for the line 18.
  8. lines 71-72: and the product was approved
  9. line 72: tests instead of testings
  10. line 81: the French market
  11. line 87: of the fermentation process
  12. line 98: a humidity of
  13. line 108: (mid-dose), or
  14. line 118: the determination
  15. line 119: erase ïn"
  16. line 120: included
  17. line 141: the aorta
  18. line 145: a fixative
  19. line 146: a thickness
  20. line 149: the lower dose groups
  21. line 178: of ME
  22. line 180: in hemoglobin
  23. line 183: erase "with"
  24. line 192: of treatment
  25. line 199: and are not dose-dependent
  26. line 231: a cyst
  27. line 233: minimally increased
  28. line 243: a difficult job
  29. line 244: the cultivation
  30. line 245: a large volume
  31. lines 238-242: Again the same as in the introduction section. Change this part by giving it from another perspective; mycelium is not tasty and can not substitute fruit bodies based on aroma and taste.
  32. line 249: it possesses
  33. line 250: shown
  34. line 261: carbohydrates
  35. Lines 261-263: Be careful with those comments; carbohydrates from mushrooms differ greatly ( in their quality) from those from other sources. Merely giving their amount in the material does not represent its value.
  36. Write more elaborately about fat composition since mycelia obtained in this research had a really big amount of it! And that is not typical for fruit bodies. It would greatly improve this work if the lipid profile (fatty acids) can be added! Mushrooms are generally rich in unsaturated fatty acids and if their profile in morels is "good" it can be a bonus benefit for their nutritive composition.
  37. The nutritive composition (based on the authors results) must be more deeply explained since it is an integral part and aim of this study.
  38. The discussion section is almost a total repeat of the results section. There is no actuals discussion! What about other studies, other mushroom species, or novel food that has a similar nutrition profile? How are the safety tests results connected with ME nutritional profile?
  39. Line 307: NOEL

Author Response

Although the study "Nutrition and safety of Morchella esculenta mycelia produced by fermentation in bioreactors" presents an important and interesting topic it needs major changes prior to publishing. Its major flaws are inadequate introduction (background/perspective) and lack of proper discussion. Minor English corrections should be done as well. Noone eats mycelia as a substitute for mushrooms (fruit body) due to its lack of flavor. Also, do we know the price of mycelia (amount which would be enough for one proper meal) in comparison with fruit bodies?

Authors’ response:
The reviewer’s comments and suggestions are greatly appreciated. The dry mushroom is around 360 US dollars per kg while the dry mycelium is around 115 US dollars in Taiwan.

  1. The title should be changed to a more precise one: Nutritional profile and animal-tested safety  of Morchella esculenta mycelia produced by fermentation in bioreactor

Authors’ response:
The reviewer’s comments and suggestions are greatly appreciated. We updated the information as advised in the revised manuscript (Title).

  1. line 16: the cultivation

Authors’ response:
We thank the reviewer’s comments and suggestions. We updated the information as advised in the revised manuscript (line 20).

  1. line 18: "to prove its use" is too vague, make it more clear, for example, to prove it is safe for human consumption while providing high-quality nutrients.

Authors’ response:
The reviewer’s comments and suggestions are greatly appreciated. We updated the information as advised in the revised manuscript (line 22-3).

  1. line 19: of oral toxicity

Authors’ response:
We thank the reviewer’s comments and suggestions. We updated the information as advised in the revised manuscript (line 24).

  1. line 24: delete one "rats"

Authors’ response:
The reviewer’s comments and suggestions are greatly appreciated. We updated the information as advised in the revised manuscript (line 28).

  1. lines 40-42: It seems that the appreciation of morels justifies the need for their cultivation. The authors are suggesting mycelium cultivation. The final product-mycelia, is completely different from the sensory aspect and people praise them mainly because of their aroma and flavor (if mushrooms are taken as food); just consider the truffles! So, if high price and demand are to be the reason for an alternative source of this food then the real reason for their appreciation must be taken into account. Thus, this part of your paper needs to be rewritten from the nutritive perspective or medicinal one, not flavor and delicacy-based. Mycelium lacks any of that. Or, if this stance is to be kept the research needs sensory evaluation of bioreactor obtained mycelia.

Authors’ response:
We thank the reviewer’s comments and suggestions. We have rewritten the paragraph from the nutritive perspective. For better clarification, we have added the following sentences “In Japan, Malaysia, India and Parkistan, morels function as natural aphrodisiacs based on ethnomycological studies. In current scientific research, morel mushrooms were found to contain profound anti-tumor, immunoregulatory, anti-inflammatory and antioxidant activities. As morels have enormous health benefits, the market demand for morels has increased accordingly.” in the revised manuscript (lines 44-52).

  1. lines 68-70: the sample as for the line 18.
    Authors’ response:
    The reviewer’s comments and suggestions are greatly appreciated. We updated the information as advised in the revised manuscript (line 81).

  1. lines 71-72: and the product was approved

Authors’ response:
We thank the reviewer’s comments and suggestions. We updated the information as advised in the revised manuscript (line 83-4).

  1. line 72: tests instead of testings
    Authors’ response:
    The reviewer’s comments and suggestions are greatly appreciated. We updated the information as advised in the revised manuscript (line 84).

  1. line 81: the French market
    Authors’ response:
    We thank the reviewer’s comments and suggestions. We updated the information as advised in the revised manuscript (line 93).

  1. line 87: of the fermentation process
    Authors’ response:
    The reviewer’s comments and suggestions are greatly appreciated. We updated the information as advised in the revised manuscript (line 99).

  1. line 98: a humidity of
    Authors’ response:
    We thank the reviewer’s comments and suggestions. We updated the information as advised in the revised manuscript (line 112).

  1. line 108: (mid-dose), or
    Authors’ response:
    The reviewer’s comments and suggestions are greatly appreciated. We updated the information as advised in the revised manuscript (line 123).

  1. line 118: the determination
    Authors’ response:
    We thank the reviewer’s comments and suggestions. We updated the information as advised in the revised manuscript (line 135).

  1. line 119: erase ïn"
    Authors’ response:
    The reviewer’s comments and suggestions are greatly appreciated. We updated the information as advised in the revised manuscript (line 136).

  1. line 120: included
    Authors’ response:
    We thank the reviewer’s comments and suggestions. We updated the information as advised in the revised manuscript (line 137).

  1. line 141: the aorta
    Authors’ response:
    The reviewer’s comments and suggestions are greatly appreciated. We updated the information as advised in the revised manuscript (line 158).

  1. line 145: a fixative
    Authors’ response:
    We thank the reviewer’s comments and suggestions. We updated the information as advised in the revised manuscript (line 162).

  1. line 146: a thickness
    Authors’ response:
    The reviewer’s comments and suggestions are greatly appreciated. We updated the information as advised in the revised manuscript (line 163).

  1. line 149: the lower dose groups
    Authors’ response:
    We thank the reviewer’s comments and suggestions. We updated the information as advised in the revised manuscript (line 167).

  1. line 178: of ME
    Authors’ response:
    The reviewer’s comments and suggestions are greatly appreciated. We updated the information as advised in the revised manuscript (line 195).

  1. line 180: in hemoglobin
    Authors’ response:
    We thank the reviewer’s comments and suggestions. We updated the information as advised in the revised manuscript (line 197).

  1. line 183: erase "with"
    Authors’ response:
    The reviewer’s comments and suggestions are greatly appreciated. We updated the information as advised in the revised manuscript (line 200).

  1. line 192: of treatment
    Authors’ response:
    We thank the reviewer’s comments and suggestions. We updated the information as advised in the revised manuscript (line 209).

  1. line 199: and are not dose-dependent
    Authors’ response:
    The reviewer’s comments and suggestions are greatly appreciated. We updated the information as advised in the revised manuscript (line 218).

  1. line 231: a cyst
    Authors’ response:
    We thank the reviewer’s comments and suggestions. We updated the information as advised in the revised manuscript (line 251).

  1. line 233: minimally increased
    Authors’ response:
    The reviewer’s comments and suggestions are greatly appreciated. We updated the information as advised in the revised manuscript (line 253).

  2. line 243: a difficult job
    Authors’ response:
    We thank the reviewer’s comments and suggestions. We updated the information as advised in the revised manuscript (line 270).

  1. line 244: the cultivation
    Authors’ response:
    The reviewer’s comments and suggestions are greatly appreciated. We updated the information as advised in the revised manuscript (line 271).

  2. line 245: a large volume
    Authors’ response:
    We thank the reviewer’s comments and suggestions. We updated the information as advised in the revised manuscript (line 272).

  1. lines 238-242: Again the same as in the introduction section. Change this part by giving it from another perspective; mycelium is not tasty and can not substitute fruit bodies based on aroma and taste.

Authors’ response:
We thank the reviewer’s comments and suggestions. We have rewritten the paragraph from the nutritive perspective. For better clarification, we have added the following sentences “According to a recent study, M. esculenta contained all the important nutrients including carbohydrates, proteins, polyunsaturated fatty acids, and several bioactive com-pounds such as polysaccharides, organic acids, polyphenolic compounds and tocopherols.  These substances, when extracted from M. esculenta, are promising for anti-oxidants, immunomodulation, anti-cancer and anti-inflammatory applications.” in the revised manuscript (lines 264-9).

  1. line 249: it possesses
    Authors’ response:
    The reviewer’s comments and suggestions are greatly appreciated. We updated the information as advised in the revised manuscript (line 276).

  2. line 250: shown
    Authors’ response:
    We thank the reviewer’s comments and suggestions. We updated the information as advised in the revised manuscript (line 277).

  1. line 261: carbohydrates
    Authors’ response:
    We thank the reviewer’s comments and suggestions. We updated the information as advised in the revised manuscript (line 285).

  2. Lines 261-263: Be careful with those comments; carbohydrates from mushrooms differ greatly ( in their quality) from those from other sources. Merely giving their amount in the material does not represent its value.

Authors’ response:
The reviewer’s comments and suggestions are greatly appreciated. We have deleted these comments in the revised manuscript (line 297-301).

  1. Write more elaborately about fat composition since mycelia obtained in this research had a really big amount of it! And that is not typical for fruit bodies. It would greatly improve this work if the lipid profile (fatty acids) can be added! Mushrooms are generally rich in unsaturated fatty acids and if their profile in morels is "good" it can be a bonus benefit for their nutritive composition.

Authors’ response:
We thank the reviewer’s comments and suggestions. We have elaborated more for the fat composition. For better clarification, we have added “Mushrooms are also healthy sources of essential fatty acids such as linoleic, oleic, and linolenic, which cannot be directly synthesized in the human body but is required for health. Compared to the fat composition of 39 common and popular mush-room species that ranged from 0.2 % to 8%, the fermented M. esculenta mycelia in this study had 17.17 % crude fat, which is not typical for fruiting bodies. After subdivided into saturated and unsaturated fats, there were 14.6 % saturated fat, 0 % trans-fat and 85.4 % unsaturated fats in the fermented M. esculenta mycelia (data not shown). Consistent with a previous study, unsaturated fatty acids were higher than saturated fatty acids and linoleic, oleic, and palmitic were the predominant fatty acids in the lipid content of M. esculenta. According to a previous study, unsaturated fats are an important part of a healthy diet as these fats help reduce the risk of heart disease and lower cholesterol levels when they replace saturated fats in the diet. As human metabolism cannot synthesize linoleic and linolenic acids as they lack enzymes for ω-3 desaturation, this suggested that fermented M. esculenta mycelia can be an important source of essential fat acids for a human health diet.” in the revised manuscript (lines 302-316).

  1. The nutritive composition (based on the authors results) must be more deeply explained since it is an integral part and aim of this study.

Authors’ response:
We thank the reviewer’s comments and suggestions. We have elaborated more for the protein (lines 291-7) and fat composition (lines 30-316) in the revised manuscript.

  1. The discussion section is almost a total repeat of the results section. There is no actuals discussion! What about other studies, other mushroom species, or novel food that has a similar nutrition profile? How are the safety tests results connected with ME nutritional profile?

Authors’ response:
We thank the reviewer’s comments and suggestions. We have compared the nutritional profile with other mushroom species. For better clarification, we have added “When compared to other 19 studied culinary-medicinal mushroom mycelia in another study, the fermented M. esculenta mycelia in this study has the highest protein content while the carbohydrate, fat and ash contents were lower when compared to mycelia of Ganoderma tsugae, Grifola frondosa, Hypsizygus marmoreus white and Pleurotus citrinopileatus.” in the revised manuscript (lines 286-90). Moreover, to connect nutritional profile with the safety study, we have added the following sentences “As the nutritional profile of M. esculenta mycelium were not inferior to those of wild M. esculenta fruiting bodies, suggested that fermented M. esculenta mycelium may be a better alternative for healthcare product development. However, mushroom mycelium and fruiting bodies are two distinct parts of the fungal organism and hence it is important to take its safety and edibility as a food ingredient into considerations using a 90-days oral subchronic toxicity test.” in the revised manuscript (lines 317-22).

  1. Line 307: NOEL
    Authors’ response:
    We thank the reviewer’s comments and suggestions. We updated the information as advised in the revised manuscript (line 372-3).

Reviewer 2 Report

Please see on comments file.

Author Response

  1. What makes morel mushrooms famous is their delicious taste. Can the mycelia cultivated byn submerge techniques match the taste of the fruiting bodies?

Authors’ response:
We thank the reviewer’s comments and suggestions. We believe that mycelia cultivated by submerge techniques match the nutrition of the fruiting bodies. For better clarification, we have rephrased the sentence “As M. esculenta mycelia has various health promoting and medicinal properties, it could be used as raw materials in formulation and development of new functional foods for health-conscious consumers.” in the revised manuscript (lines 77-9).

  1. To clarify the “similarity” add the proportion(value) of macronutrients mentioned by the authors in Table 1 or elsewhere. This is important to know the correlation between proximate analysis results and DRI recommendation

Authors’ response:
We thank the reviewer’s comments and suggestions. We updated the information as advised in the revised manuscript (Table 1).

  1. The reference article (reference no 18) that isreferred to explain ME sample preparation before being administered to experimental animals do not appear to be available in online journals, or perhaps in Chinese-language journals. It is recommended that the preparation of samples before being given to experimental animals should described in this manuscript.

Authors’ response:
We thank the reviewer’s comments and suggestions. We have described the sample preparation for reference 18 (now changed to reference 20). For better clarification, we have rephrased the following sentence “The dose levels were determined based on the results of our previous teratotoxic assessment, which used the same method in this study to prepare ME samples and showed no adverse effect when up to 3000 mg/kg bw/day of ME was given to female pregnant rats during the major embryonic organogenesis period (days 6-15).” In the revised manuscript (lines 125-7).

  1. Clearly identify the location (city and country) of the mentioned French market

Authors’ response:
We thank the reviewer’s comments and suggestions. For better clarification, we have updated the city and country of the French market (Paris, France) in the revised manuscript (lines 93).

  1. Provide the AOAC code for each method of analysis for moisture, ash, crude fiber and crude protein content

Authors’ response:
We thank the reviewer’s comments and suggestions. For better clarification, we have updated AOAC code for each method of analysis “moisture (method 925.10), ash (method 923.03), crude fiber (method 962.09), crude fat (method 920.39) and crude protein content (method 979.06)” in the revised manuscript (lines 104-6).

  1. Probably the "ethnics" should be "ethics". It is recommended that the year of ethical approval is also displayed.

Authors’ response:
We thank the reviewer’s comments and suggestions. For better clarification, we have changed the word from "ethnics" to "ethics" and updated the year of ethical approval in the revised manuscript (lines 109 and 117, respectively).

  1. Add information on type of gavage needle the authors used

Authors’ response:
We thank the reviewer’s comments and suggestions. For better clarification, we have added the type gavage needle (a straight, stainless steel ball-tipped feeding needle) in the revised manuscript (lines 121-2).

  1. What substance is used for negative control administration?

Authors’ response:
We thank the reviewer’s comments and suggestions. For better clarification, we have added “sterile distilled water as negative control” in the revised manuscript (line 122).

  1. How long did it take for the fixation time? Same time with the one for other additional tissues (line 145)?

Authors’ response:
We thank the reviewer’s comments and suggestions. For better clarification, we have added “for 24 h” in the revised manuscript (lines 158 and 162).

  1. Please avoid redundant writing of analysis results in sentences and Table 1

Authors’ response:
We thank the reviewer’s comments and suggestions. For better clarification, we have deleted the following sentence “The mycelia contained 4.20±0.49 % moisture, 0.32± 0.07% total ash, 17.17±0.07 % crude lipid, 39.35±0.35 % crude protein, 38.96±4.60 % carbohydrates and 467.77±0.21 kcal/100 g energy.” in the revised manuscript (lines 174-6).

  1. However, it is still observable that High dose treatments (both for male and female rats) resulted in lower body weight, especially when they are compared with Control and Mid dose groups. How would the author expalin this?

Authors’ response:
We thank the reviewer’s comments and suggestions. For better clarification, we have added the following sentence “This significant decrease may be caused by the 39.35 % crude protein within the ME mycelia. According to previous studies, protein-rich loads can contribute to longer-lasting satiety, decrease subsequent energy intake and lead to lower body weight.” in the revised manuscript (lines 329-32).

  1. Does this phenomenon cause a lower body weight? See comment No. 6.

Authors’ response:
We thank the reviewer’s comments and suggestions. For better clarification, we have added the following sentence “This significant decrease may be caused by the 39.35 % crude protein within the ME mycelia. According to previous studies, protein-rich loads can contribute to longer-lasting satiety, decrease subsequent energy intake and lead to lower body weight.” in the revised manuscript (lines 329-32).

  1. These phenomenon (increase in haemoglobin and RBC) could be elaborated more rather than just saying (line 184) “to be unrelated”. In malerats, Will a higher production of Red Blood Cells affect a higher haemoglobin number?

Authors’ response:
We thank the reviewer’s comments and suggestions. For better clarification, we have added the following sentence “A significant increase in hemoglobin values and a similar trend for the RBCs, however, were found in the male mid- and high-dose treated groups, which may be due to strong anti-oxidant effect of ME mycelia to prevent the destruction of RBCs from free radical formation” in the revised manuscript (lines 336-40).

  1. Please fix the table so that the columns can be seen more clearly

Authors’ response:
We thank the reviewer’s comments and suggestions. For better clarification, tables 2, 3 and 4 have been fixed.

  1. The High dose treatment results of ALT in Male Rats were quite deviating and not following the trend. For example (in ALT), Control (50.2) is higher than Low (42.3) and Mid (39.6) but it is lower than High dose (60.9). Any explanation?

Authors’ response:
We thank the reviewer’s comments and suggestions. For better clarification, we have added the following sentence “Both male and female rats administered with low-dose and medium-dose ME mycelia treatment exhibited decreased AST and ALT, suggesting that the lower doses of ME mycelia showed obvious protective effects on liver function. Although elevated serum ALT pointed to a potential risk of liver injury, the changes that occurred at the clinical chemistry levels did not correspond in the histopathological examination, indicating that ME mycelia had little effect on tissue levels.” in the revised manuscript (lines 351-6).

  1. Then, it is important to provide the values of normal range for previously mentioned parameters (line 193 – 197).

Authors’ response:
We thank the reviewer’s comments and suggestions. For better clarification, we have added the normal range “(ALT=18.72-85.05 U/L; Calcium=11.20-13.02 mg/dL; Phosphorus= 9.49-13.01 mg/dL; Sodium=142.47-150.41 meq/L and BUN=11.8-20.0 mg/dL)” in the revised manuscript (lines 215-8).

  1. Table 4 could be simplified. Why should the auhors always use (/10) if they are all the same?

Authors’ response:
We thank the reviewer’s comments and suggestions. For better clarification, we have deleted (/10) in the revised manuscript (Table 4).

  1. What are the units for the organs?

Authors’ response:
We thank the reviewer’s comments and suggestions. For better clarification, we have added (g) in the titles of table 5 and 6 of the revised manuscript.

  1. Provide the normal range for the mentined parameters. Authors may add this information in Table 5 or in body text (between line 213 –221)

Authors’ response:
We thank the reviewer’s comments and suggestions. For better clarification, we have added the normal range “(male absolute heart weight: 1.544-1.916 g; female absolute spleen weight: 0.43-0.78 g and relative ovary weight: 0.024-0.044 g)” in the revised manuscript (lines 238-40).

  1. So, if it is not due to ME mycelia, what are the main factors causing these lesions? Authors should elaborate this phenomenon rather than just expecting it is a sporadic phenomenon.

Authors’ response:
We thank the reviewer’s comments and suggestions. For better clarification, we have added the following sentence “According to a previous study, old rats will have a subset of lesions as part of the progressive decline in organ function.” in the revised manuscript (lines 254-6).

  1. What is meant by incidental? Does it mean the mentioned data not reliable? It would be better if authors could provide reference that could support the authors’ statement.

Authors’ response:
We thank the reviewer’s comments and suggestions. For better clarification, we have rephrased the following sentence “although significant differences in absolute weights were discovered in the heart of high-dose treated males and in the spleen of mid-dose treated females, these weights are within the normal range. Moreover, no obvious lesions were found in further histopathological examination and hence these changes could be considered of no clinical significance.” in the revised manuscript (lines 360-4).

Round 2

Reviewer 1 Report

some minor grammar mistakes (e.g., where instead of was). The paper should be checked for grammar.

Author Response

some minor grammar mistakes (e.g., where instead of was). The paper should be checked for grammar.

Author's responses:
We sincerely appreciate the reviewer’s comments. We have polished this 
manuscript so we hope it now matches the journal standard.